# Improving the Convergence of Dynamic NeRFs via Optimal Transport

**Sameera Ramasinghe**[*], **Violetta Shevchenko**[*], **Gil Avraham, Hisham Husain, Anton van den Hengel**
Amazon, Australia

## Abstract

Synthesizing novel views for dynamic scenes from a collection of RGB inputs poses significant challenges due to the inherent under-constrained nature of the problem. To mitigate this ill-posedness, practitioners in the field of neural radiance fields (NeRF) often resort to the adoption of intricate geometric regularization techniques, including scene flow, depth estimation, or learned perceptual similarity. While these geometric cues have demonstrated their effectiveness, their incorporation leads to evaluation of computationally expensive off-the-shelf models, introducing substantial computational overhead into the pipeline. Moreover, seamlessly integrating such modules into diverse dynamic NeRF models can be a non-trivial task, hindering their utilization in an architecture-agnostic manner. In this paper, we propose a theoretically grounded, lightweight regularizer by treating the dynamics of a time-varying scene as a low-frequency change of a probability distribution of the light intensity. We constrain the dynamics of this distribution using optimal transport (OT) and provide error bounds under reasonable assumptions. Our regularization is learning-free, architecture agnostic, and can be implemented with just a few lines of code. Finally, we demonstrate the practical efficacy of our regularizer across state-of-the-art architectures. Our code is at `https://github.com/samgregoost/OTDNeRF/`

## 1 Introduction

Synthesizing novel views of a dynamic scene, given a sequence of monocular frames (Park et al., 2010; Gao et al., 2022) or a set of time-synchronized cameras (Li et al., 2022), has recently gained significant traction in the field (Brickwedde et al., 2019b; Park et al., 2021a;b; Pumarola et al., 2021; Cai et al., 2022; Li et al., 2023a). The concept of novel-view synthesis originates from the realm of image-based rendering (IBR) (Gortler et al., 1996), where the objective is to accurately aggregate light intensity information from a dense collection of images to generate a desired novel view. One of the most noteworthy recent developments in deep learning for IBR is NeRF (Mildenhall et al., 2021), which can replicate the capabilities of a meticulously designed array of cameras using state-of-the-art Structure from Motion (SfM) algorithms (Schonberger & Frahm, 2016).

Despite NeRF's impressive results and robust performance, numerous recent works have embarked on the challenging task of extending NeRF to dynamic scenes (Pumarola et al., 2021; Li et al., 2023b; Park et al., 2021b; Fang et al., 2022; Ramasinghe et al., 2023). However, transitioning from static to dynamic settings presents a non-trivial and challenging problem replete with complexities (Deng et al., 2022). Notably, disentangling camera motion from object motion inherently constitutes an ill-posed problem, further complicated by variations in lighting, non-rigid object deformations, and their interplay. To address these additional complexities, explicit regularizations are imperative to constrain the problem. Most existing approaches that employ such regularizations rely on geometric constraints (e.g., depth losses, flow-based losses) or learned priors (e.g., maintaining consistent perceptual similarity across rendered frames over time) to guide dynamic NeRFs toward plausible solutions.

However, these regularization techniques often entail adding additional deep networks on top of already bulky NeRF models or involve expensive preprocessing steps. For instance, perceptual

---

[*]Authors contributed equally.

regularizers require backpropagation through pre-trained deep networks, while depth-based losses demand capturing depth information (e.g., using LIDAR) or running off-the-shelf depth estimation models during data preprocessing. Flow-based losses similarly necessitate extra preprocessing steps involving external models. Furthermore, these regularizers enforce priors that are averaged over large datasets, which may lead to domain gaps when applied to specific scenes with unique geometries. Errors in estimating geometric cues can directly propagate from external models to the NeRF optimization process.

In contrast, we introduce a straightforward plug-in regularizer that leverages statistical regularities of rendered frames from fixed camera views to constrain optimization for *unseen* views during training. Our central hypothesis is that the pixel intensity distribution of a scene, as rendered from a specific fixed camera view, should remain approximately consistent within short time intervals. However, due to object motion, pixel coordinates can change over time, making it challenging to minimize a pixel-to-pixel distance function between two rendered frames. To address this limitation, we propose to minimize a dissimilarity measure between pixel intensity *distributions* to maintain statistical regularity. One effective approach, in this vein, is to use a geometric divergence metric between distributions. Two primary candidates include optimal transport (OT) based metrics and maximum mean discrepancy (MMD), each with its own merits and downsides.

OT-based metrics exploit the geometry of the underlying ground space, making them more favourable over MMD. However, there are two key challenges: 1) solving an OT problem is generally of $O(n^3)$ complexity, making it impractical to compute in each iteration, whereas MMD is computationally efficient, since it can be expressed as a finite-dimensional inner product; and 2) rendering the entire pixel distribution of a particular frame at each iteration consumes significant memory in the NeRF framework. To address the former, we employ a sliced-Wasserstein approximator, which reduces OT computation complexity to $O(n \log n)$, significantly enhancing computational efficiency even compared to MMD. To overcome the latter, we obtain an unbiased estimator of the full pixel distribution and further refine the estimator with interpolation for unsampled pixel coordinates. Our experiments demonstrate that the sliced-Wasserstein approximator offers both faster computation and superior performance compared to MMD variants. We also show that the Wasserstein distance in our setting converges at competitive rates of $O(n^{-1/2})$ with poly-logarithmic dependence on the number of observed values in the image. Importantly, our regularizer can be seamlessly integrated into any dynamic NeRF architecture without altering the existing model. Our method assumes no prior knowledge about the scene and adapts dynamically at training. Empirically, we demonstrate that the proposed regularizer enhances the performance of various dynamic NeRFs. Moreover, ablation studies illustrate that our simple regularizer surpasses existing approaches that rely on resource-intensive deep models or costly preprocessing steps.

## 2 RELATED WORK

D-NeRF Pumarola et al. (2021), demonstrated that in the context of a dynamic object-centric scene, a canonical representation assumption could be made. This assumption allows for the optimization of implicit motion and ray interactions with respect to a reference frame while treating neighboring frames as small deformations. This paradigm involves the coordination of two Multilayer Perceptrons (MLPs): the first one is responsible for deforming rays from the required view to the canonical view, essentially inferring the delta ray bend; the second MLP applies the delta ray-bending from the canonical reference frame to the desired view. The ray-bending paradigm with respect to a canonical frame is effective, but, inherently under-constrained. Works by Tretschk et al. (2021), Park et al. (2021a), and Park et al. (2021b) introduced several regularizations to constrain the allowed ray deformations. NR-NeRF by Tretschk et al. (2021) regularized the magnitude of the ray offsets, effectively limiting their deformation to reasonable changes, assuming that the motion is not abrupt. However, due to the absence of explicit geometry, this regularization is applied to the entire deformation volume, including occluded and unseen areas, requiring specific scheduling to stabilize. Similarly, Park et al. (2021a) employed elastic regularization, which enforces constraints on the singular values of the Jacobian of the deformation MLP, driving them towards zero. By doing so, they achieve a uniform scale of changes in the deformation directions.

Nevertheless, it was observed that elastic regularization faced challenges in scenarios with complex topological deformations. HyperNeRF (Park et al., 2021b), addressed discontinuities in the defor-

mation field, particularly when changes in surface connectivity occur (e.g., opening a mouth, cutting a lemon) or transient objects appear (e.g., fire or flame). To address this, HyperNeRF models frames as the zero-level set of a higher-dimensional ambient space. Consequently, to infer novel views in practice, HyperNeRF requires accurate slicing of the ambient space and obtaining a latent representation, which is then used to model the ray-deformation. Similar to NR-NeRF, this approach deals with complexity through over-parametrization, necessitating careful scheduling during optimization. Drawing inspiration from Image-Based Rendering (IBR), researchers have also explored the use of explicit geometry, such as monocular depth estimation (Ranftl et al., 2021) and scene flow (Brickwedde et al., 2019a), as active regularization techniques for the ray-deformation operation.

## 3 METHODOLOGY

Sec. 3.1 presents a brief exposition of statistical divergence metrics and motivates our approach. Sec. 3.2 delves deeper into the proposed regularizer.

### 3.1 STATISTICAL DIVERGENCES

Let $\Sigma$ denote the standard Borel $\sigma$-algebra on $\mathbb{R}^d$ and $\mathcal{P}(\mathbb{R}^d)$ to denote the set of Borel probability measures (positively signed Borel measures with each $\mu \in \mathcal{P}(\mathbb{R}^d)$ satisfying $\mu(\mathbb{R}^d) = 1$. When it comes to comparing two probability distributions, there are a plethora of different options, referred to as *divergences*. A classic choice is the $f$-divergence (or the Czisar divergence (Csiszár, 1975)) which for two probability distributions $\mu, \nu \in \mathcal{P}(\mathbb{R}^d)$ is defined as $I_f(\mu : \nu) = \int_{\mathbb{R}^d} f(d\mu/d\nu)d\nu$ if $\mu$ is absolutely continuous with respect to $\nu$ and $\infty$ otherwise where $f : (-\infty, \infty] \to \mathbb{R}$ is a convex lower semicontinuous function such that $f(1) = 0$. In particular, this divergence is a generalization of the well-known Kullback-Leibler (KL) divergence for the choice of $f(t) = t \log t$. Despite the widespread use of $f$-divergences, they are computationally intractable in the setting when only samples are available and require absolute continuity between $\mu$ and $\nu$ which often surfaces as a stringent requirement.

Natural remedies to this issue is to consider Integral Probability Metrics (IPM), which is the motivation of works in other domains such as generative models (Mroueh & Sercu, 2017) and distributional robustness (Staib & Jegelka, 2019; Husain, 2020). There are two prevalent choices of IPMs that are used: the (kernel) Maximum Mean Discrepancy (MMD) (Gretton et al., 2012) and the $p$-Wasserstein distance (Villani et al., 2009). The MMD is typically favoured for its computational tractability as it can be computed in $O(n^2)$ time where $n$ is the number of samples however relies on an appropriate choice of characteristic kernel function. On the other hand the $p$-Wasserstein distance utilizes a ground metric on the support of the distributions and characterizes a divergence at the statistical level. More formally, the $p$-Wasserstein distance between two probability measures $\mu, \nu \in \mathcal{P}(\mathbb{R}^d)$ is

$$W_p^p(\mu_1, \mu_2) = \inf_{\upsilon \in \Pi(\mu,\nu)} \int_{\mathbb{R}^d \times \mathbb{R}^d} ||x - y||^p d\upsilon(x, y), \tag{1}$$

where $\Pi(\mu, \nu)$ is the set of all *couplings*: joint probability measures on $\mathbb{R}^d \times \mathbb{R}^d$ with marginals $\mu$ and $\nu$, and $|| \cdot ||$ is the Euclidean norm. The Wasserstein distances solve for a mass transportation problem by finding the minimal coupling $\upsilon$ and as such, alleviate issues with $f$-divergence such as non-overlapping support. However Wasserstein distances remain analytically intractable for all but a few cases. One straightforward way to solve the discrete OT problems is to use linear programming based algorithms such as the Hungarian method Kuhn (1955), the auction algorithm Bertsekas (1988) and the network simplex Waissi (1994), which are typically numerically robust. Unfortunately, these methods pay a large price in algorithmic complexity, especially the memory requirements for solving larger problems. However, in the 1-dimensional setting, we can compute the $p$-Wasserstein distance in $O(n \log n)$ time. Consider two finitely support distributions: $\hat{\mu} = \frac{1}{n} \sum_{i=1}^n \delta_{x_i}, \hat{\nu} = \frac{1}{n} \sum_{i=1}^n \delta_{y_i}$ where $x_i, y_i \in \mathbb{R}$ where $\delta_z(z')$ is the *dirac* delta function that corresponds to 1 if $z' = z$ and 0 otherwise. In this case, the $p$-Wasserstein distance can be computed with $W_p^p(\hat{\mu}, \hat{\nu})^p = \frac{1}{n} \sum_{i=1}^n |\tilde{x}_i - \tilde{y}_i|^p$, where $\{\tilde{x}_i\}$ and $\{\tilde{y}_i\}$ are sorted variants of $\{x_i\}$ and $\{y_i\}$ such that $\tilde{x}_1 < \cdots < \tilde{x}_n$ and $\tilde{y}_1 < \cdots < \tilde{y}_n$. This problem can be solved at $O(n \log n)$ cost, which is the sorting cost. This property of the Wasserstein distance has been heavily leveraged in many practical applications, using the Sliced-Wasserstein (SW) distance Rabin et al. (2012); Bonneel et al. (2015); Avraham et al. (2019). More specifically, Sliced Wasserstein distance provides a

practical univariate approximation to the multi-dimensional case, with robust theoretical guarantees. Let $\mathbb{S}^{d-1}$ be the $d$-dimensional unit sphere and $\gamma$ the uniform distribution on the surface of $\mathbb{S}^{d-1}$. For $\theta \in \mathbb{S}^{d-1}$, $\theta^* : \mathbb{R}^d \to \mathbb{R}$ denotes the linear form $x \to \langle x, \theta \rangle$ with $\langle \cdot, \cdot \rangle$ the Euclidean inner-product. Then, SW of order $p$ is

$$SW(\mu_1, \mu_2) = \int_{\mathbb{S}^{d-1}} W_p(\theta^*_{\#}\mu_1, \theta^*_{\#}\mu_2) d\gamma(\theta), \tag{2}$$

where for any measurable function $f : \mathbb{R}^d \to \mathbb{R}$ and $\xi \in \mathcal{P}(\mathbb{R}^d)$ is the push-forward measure of $\xi$ by $f$: for any Borel set $A$ in the $\sigma$-algebra of $\mathbb{R}$, $f_{\#}\eta(A) = \eta(f^{-1}A)$, where $f^{-1}(A) = \{x \in \mathbb{R}^d : f(x) \in A\}$.

In this work, we treat images as distributions with finite support. For a grayscale image $I \in \mathbb{R}^{W \times H \times 1}$ of size $W \times H$, let $\Omega \subset \mathbb{R}$ denote the set of observed values in the true image. We use $\mathcal{P}(\{\mathbf{r}\})$ to denote the distribution from images whose observed values are $\{\mathbf{r}\}$. Then, we present the following theorem.

**Theorem 1** *For any two sets of samples $\mathcal{P}(\{\hat{\mathbf{r}}^{t_1}\})$ and $\mathcal{P}(\{\hat{\mathbf{r}}^{t_2}\})$ drawn i.i.d from t base distributions $P(\{\mathbf{r}^{t_1}\})$ and $P(\{\mathbf{r}^{t_2}\})$ supported on a finite set of values $\Omega$, the following holds*

$$\left| W_1\left(\mathcal{P}(\{\hat{\mathbf{r}}^{t_1}\}), \mathcal{P}(\{\hat{\mathbf{r}}^{t_2}\})\right) - W_1(\mathcal{P}(\{\mathbf{r}^{t_1}\}), \mathcal{P}(\{\mathbf{r}^{t_2}\})) \right| \leq 4\sqrt{\frac{|\Omega|^2}{2n} \log\left(\frac{2|\Omega|}{\delta}\right)}, \tag{3}$$

*with probability at least $1 - \delta$.*

(Proof in Appendix). It should be noted here that the rate of convergence depends on the size of $|\Omega|$. We can see from above that the 1-Wasserstein distance from empirical samples at two time-stamps converges competitively. In practice, we observe that the performance increases with $n$ (Table 7), possibly due to the lower error in OT computation as predicted by the above result.

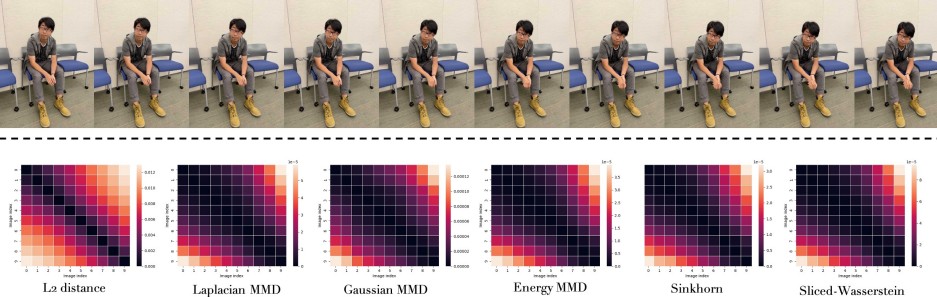

Figure 1: **An example illustration of different divergence metrics over temporarily varying pixel distributions captured from a fixed camera.** *Top row*: An image sequence extracted from the *space-out* test sequence of the *iPhone dataset*. *Bottom row*: The heatmaps indicate the metric distance between the corresponding images. As shown, pixel-to-pixel $L_2$ loss does not reasonably capture the similarity between images as the objects in the scene are dynamic. In comparison, geometric distances better capture the similarity of pixel distributions, where sliced-Wasserstein distance performs best (zoom in for a better view). We notice that this slight enhancement results in a notable boost in performance within the context of the more intricate dynamic NeRF scenario, potentially due to its ability to effectively handle subtle variations in lighting conditions.

## 3.2 OPTIMAL TRANSPORT FOR IMPROVING THE CONVERGENCE OF DYNAMIC NERFS

We consider the scenario in which the provided data consists solely of a series of snapshots and corresponding camera poses of a dynamic scene, captured by a single moving camera. This problem is highly underconstrained, as it provides only one observation from a specific camera pose at any given timestamp. Consequently, the NeRF model can potentially converge to degenerate solutions, violating multi-view consistency, unless explicit regularization measures are imposed.

Notably, we observe that for a scene with reasonably smooth temporal dynamics, the pixel intensity distributions when rendered from a fixed camera pose should remain similar within a short time interval. However, the pixel coordinates change over time due to object movements. For example,

consider the pixel distribution $\mathcal{P}(\mathbf{r}_{t,p})$ of a rendered image from a camera pose $p$ at time $t$, where $\mathbf{r}_{t,p}$ represents RGB intensities. Within a small time interval we assume $\Delta t$, $\mathcal{P}(\mathbf{r}_{t,p}) \approx \mathcal{P}(\mathbf{r}_{t+\Delta t,p})$ [1]. Therefore, the minimization of $\mathcal{D}(\mathcal{P}(\mathbf{r}_{t,p}), \mathcal{P}(\mathbf{r}_{t+\Delta t,p}))$, where $\mathcal{D}(\cdot, \cdot)$ is a suitable divergence metric that remains invariant with respect to pixel coordinates, should assist the model in converging to a more robust solution. In fact, we have found that in real-world scenes, this approach effectively regularizes the training of dynamic NeRFs on-the-fly, resulting in improved 3D reconstructions.

Fig. 1 is a toy example that illustrates the validity of this hypothesis. Note that as the person moves, the $L_2$ distance rapidly diverges from zero as the indices move away from the diagonal of the heatmap. In comparison, geometric losses are able to maintain a low metric distance across a larger area. We observe that the better performance of slice-Wasserein metric translates to a significant performance improvement in the more complex dynamic NeRF setting, possibly owing to its robustness to subtle light changes (see Sec. 4). Interestingly, we observe that sliced-Wasserstein distance performs better even compared to the Sinkhorn divergence which is an efficient approximation to the Wasserstein distance and can be considered as an interpolation between OT and MMD (Feydy et al., 2019). This particular behaviour allows us to enjoy both faster estimations ($O(n \log n)$ as opposed to $O(n^2)$ complexity of MMD) and better convergence with sliced-Wasserstein distance.

One critical obstacle that impedes the regularization of dynamic NeRFs using rendered images as inputs is their computational complexity. Rendering a complete projection of a 3D scene from a given camera angle involves tracing a ray for each pixel and densely sampling the density and color fields in 3D space. This necessitates the exhaustive evaluation of a NeRF backbone (typically an MLP or a dense feature grid) for each rendering, often demanding compute power that is infeasible. Consequently, methods utilizing image statistics for regularization are compelled to resort to using small patch renderings for this purpose, which frequently provide inadequate approximations of the global statistics of the entire projection. As these regularization methods require well-structured pixel information as inputs, rendering sparse pixel samples to obtain better global statistics is not a viable option (e.g., calculating LPIPS similarity between randomly sampled pixels from images lacks meaning). In contrast, our regularization method is based on reducing the OT cost between pixel distributions, and it is not hindered by the aforementioned obstacle. Instead, we can obtain an unbiased estimator of the true population by rendering pixel values at random coordinates, enabling us to efficiently regularize the scene with theoretical guarantees and manageable computational requirements. Fig. 2 illustrates our method graphically.

Further, we observed that interpolating to unsampled pixel coordinates leads to better performance. Therefore, we interpolate to the full image resolution after rendering random pixels before computing the sliced-Wasserstein distance. It is important to note that this only minimally affects the computational overhead since we are not actually rendering the missing pixels. We conducted experiments with more complex interpolation kernels; however, bilinear

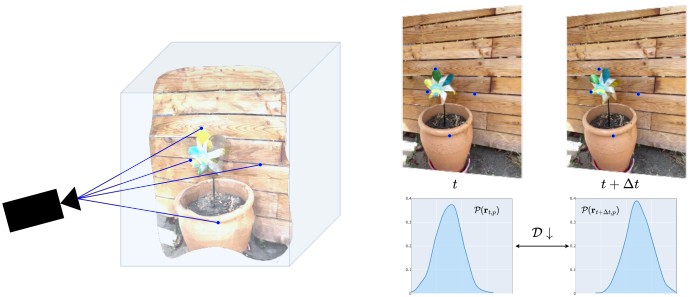

Figure 2: **A graphical demonstration of our approach**. We minimize the OT distance $\mathcal{D}$ between the pixel intensity distributions $\mathcal{P}(\mathbf{r}_{t,p})$ and $\mathcal{P}(\mathbf{r}_{t+\Delta t,p})$, where $\mathbf{r}_{t,p}$ is the set of pixels rendered from camera $p$ at time $t$.

interpolation with soft smoothing yielded the best results. For an ablation, please see Table 3. Our method is outlined in Algorithm 1.

---

[1]See Sec. 5 for limitations.

---

**Algorithm 1** The proposed regularization loss for each optimization iteration.

---

**Input:**

- $\{p_i\}_{i=0}^m$ - Training camera poses, $\Delta t$ - Time interval, $W$ - Width of rendered images, $H$ - Height of rendered images, $\beta$ - Scalar loss weight, $k$ - Number of random vectors used for sliced-Wasserstein distance, $d$ - Number of pixels sampled for computing sliced-Wasserstein distance, $L_{photo}$ - The usual photometric loss of dynamic NeRF

**Output:**

- $L_{total}$ - The result of the algorithm.

1: Randomly sample a camera pose: $p_i$
2: Sample $t_1 \sim U(0, 1)$ and $\hat{t} \sim U(0, \Delta t)$
3: Calculate $t_2 = t_1 + \hat{t}$
4: Sample 2D coordinates: $\{(x_1, y_1), (x_2, y_2), \ldots, (x_d, y_d)\} \sim U(0, W) \times U(0, H)$
5: Render pixels at $t_1$ and $t_2$ from $p_i$ at sampled coordinates: $\mathbf{R}_1, \mathbf{R}_2$        $\triangleright \mathbf{R}_1, \mathbf{R}_2 \in \mathbb{R}^{d \times 3}$
6: Sample vectors: $\mathbf{u}_1, \mathbf{u}_2, \ldots, \mathbf{u}_k \sim U(0, 1)^3$        $\triangleright \mathbf{u}_i \in \mathbb{R}^3$
7: Normalize vectors: $||\mathbf{u}_i|| = 1$
8: Create matrix: $\mathbf{U} = [\mathbf{u}_1; \mathbf{u}_2; \ldots; \mathbf{u}_k]$        $\triangleright \mathbf{U} \in \mathbb{R}^{k \times 3}$
9: Compute sliced-Wasserstein distance:

```
U_r = U.reshape(n, 1, 3)
R1_r = R1.reshape(1, d, 3)
R2_r = R2.reshape(1, d, 3)

p = (U_r * R1_r).sum(dim = 2)
q = (U_r * R2_r).sum(dim = 2)

p, _ = torch.sort(p, dim=1)
q, _ = torch.sort(q, dim=1)

RegLoss = torch.mean(torch.abs(p - q) / (1 + (p - q) ** 2))
```

10:    $L_{Total} = L_{photo} + \beta RegLoss$        $\triangleright$ Calculate total loss
11: **Return** $L_{Total}$

---

# 4 EXPERIMENTS

In this section, we empirically assess the effectiveness of the proposed regularization method by integrating it into several recent dynamic NeRF models. We used $\beta = 0.1$. Although augmenting $\Delta t$ per scene leads to improved results, we opted to fix it to $0.1$ across all scenes and datasets to better demonstrate the robustness of our method. We used $256$ or $512$ for $n$ and $2048$ or $4096$ for $d$, depending on the model size.

## 4.1 DATASETS AND BASELINES

We use the iPhone dataset proposed by Gao et al. (2022), HyperNeRF interpolation dataset, and the HyperNeRF vrig dataset (see Appendix) proposed by Park et al. (2021b) for evaluation. We choose DNeRF (Gao et al., 2021), HyperNeRF (Park et al., 2021b), Nerfies (Park et al., 2021a), TiNeuVox (Fang et al., 2022), and Hexplanes (Cao & Johnson, 2023) as baselines.

## 4.2 EVALUATION ON THE IPHONE DATASET

Many datasets used in dynamic NeRF setting do not represent practical in-the-wild capture. For instance, the datasets contain limited motions and the test sets are small, hiding issues in incorrect deformation and resulting geometry. For these reasons, Gao et al. (2022) proposed a new dataset dubbed the iPhone dataset. It consists of sequences with non-repetitive motion, from various categories such as generic objects, humans, and pets. The dataset deploys three cameras for multi-camera capture – one hand-held moving camera for training and two static cameras for evaluation. Table 1 and Fig. 3 depict quantitative and qualitative improvements by our regularization, respectively. We use the masked metrics proposed by the dataset that use covisibility masks.

## 4.3 EVALUATION ON THE HYPERNERF INTERPOLATION DATASET

HyperNeRF interpolation dataset contains six sequences with densely captured images. To make the task more challenging, we train all the models by taking only every fourth frame for training. For evaluation, we fix the first camera pose, and render frames from all the training time stamps. This setting measures how robustly the models have captured the scene dynamics when deviating from the ground truth pose, time pairs. Since we do not have the ground truth images for this eval-

| Method | Apple | | | Block | | | Paper windmill | | |
|---|---|---|---|---|---|---|---|---|---|
| | mPSNR↑ | mSSIM↑ | mLPIPS↓ | mPSNR↑ | mSSIM↑ | mLPIPS↓ | mPSNR↑ | mSSIM↑ | mLPIPS↓ |
| DNeRF | 12.635 | 0.671 | 0.669 | - | - | - | 14.628 | 0.330 | 0.375 |
| DNeRF w/ Reg | 13.454 | 0.679 | 0.649 | - | - | - | 16.383 | 0.381 | 0.351 |
| Nerfies | 15.853 | 0.709 | 0.570 | 13.860 | 0.572 | 0.523 | 12.817 | 0.223 | 0.570 |
| Nerfies w/ Reg | 17.719 | 0.762 | 0.480 | 14.398 | 0.583 | 0.527 | 16.854 | 0.342 | 0.273 |
| HyperNeRF | 15.440 | 0.704 | 0.594 | 14.535 | 0.593 | 0.502 | 13.628 | 0.256 | 0.499 |
| HyperNeRF w/ Reg | 18.056 | 0.757 | 0.464 | 16.562 | 0.639 | 0.404 | 16.946 | 0.337 | 0.265 |
| HexPlane | 16.606 | 0.712 | 0.620 | 15.888 | 0.622 | 0.542 | 16.989 | 0.356 | 0.516 |
| HexPlane w/ Reg | 17.189 | 0.718 | 0.633 | 15.893 | 0.625 | 0.537 | 17.144 | 0.365 | 0.512 |
| TiNeuVox | 9.259 | 0.682 | 0.704 | 9.793 | 0.607 | 0.502 | 11.486 | 0.289 | 0.359 |
| TiNeuVox w/ Reg | 12.561 | 0.716 | 0.596 | 11.114 | 0.625 | 0.481 | 12.233 | 0.317 | 0.349 |

| Method | Space out | | | Spin | | | Teddy | | |
|---|---|---|---|---|---|---|---|---|---|
| | mPSNR↑ | mSSIM↑ | mLPIPS↓ | mPSNR↑ | mSSIM↑ | mLPIPS↓ | mPSNR↑ | mSSIM↑ | mLPIPS↓ |
| DNeRF | 16.740 | 0.606 | 0.410 | 16.384 | 0.502 | 0.451 | - | - | - |
| DNeRF w/ Reg | 17.080 | 0.628 | 0.363 | 15.825 | 0.484 | 0.506 | - | - | - |
| Nerfies | 15.709 | 0.571 | 0.397 | 13.900 | 0.471 | 0.505 | 13.073 | 0.542 | 0.442 |
| Nerfies w/ Reg | 16.545 | 0.607 | 0.349 | 13.324 | 0.485 | 0.547 | 14.147 | 0.556 | 0.417 |
| HyperNeRF | 16.000 | 0.560 | 0.422 | 14.441 | 0.479 | 0.497 | 13.382 | 0.546 | 0.438 |
| HyperNeRF w/ Reg | 17.090 | 0.613 | 0.337 | 15.314 | 0.502 | 0.486 | 14.361 | 0.565 | 0.389 |
| HexPlane | 17.081 | 0.581 | 0.540 | 16.125 | 0.493 | 0.563 | 13.051 | 0.517 | 0.627 |
| HexPlane w/ Reg | 17.197 | 0.593 | 0.560 | 16.170 | 0.495 | 0.558 | 13.126 | 0.519 | 0.629 |
| TiNeuVox | 11.377 | 0.610 | 0.422 | 10.179 | 0.473 | 0.616 | 8.219 | 0.543 | 0.551 |
| TiNeuVox w/ Reg | 12.372 | 0.627 | 0.411 | 14.799 | 0.635 | 0.531 | 9.161 | 0.562 | 0.514 |

| Method | Wheel | | | Mean | | |
|---|---|---|---|---|---|---|
| | mPSNR↑ | mSSIM↑ | mLPIPS↓ | mPSNR↑ | mSSIM↑ | mLPIPS↓ |
| DNeRF | - | - | - | 15.097 | 0.527 | 0.476 |
| DNeRF w/ Reg | - | - | - | 15.685 | 0.543 | 0.467 |
| Nerfies | 10.109 | 0.333 | 0.511 | 13.617 | 0.489 | 0.503 |
| Nerfies w/ Reg | 11.308 | 0.392 | 0.381 | 14.899 | 0.532 | 0.425 |
| HyperNeRF | 9.741 | 0.314 | 0.537 | 13.881 | 0.493 | 0.498 |
| HyperNeRF w/ Reg | 11.863 | 0.389 | 0.386 | 15.742 | 0.543 | 0.390 |
| HexPlane | 13.044 | 0.407 | 0.577 | 15.541 | 0.527 | 0.569 |
| HexPlane w/ Reg | 13.228 | 0.434 | 0.564 | 15.707 | 0.536 | 0.570 |
| TiNeuVox | 5.801 | 0.330 | 0.604 | 9.444 | 0.504 | 0.536 |
| TiNeuVox w/ Reg | 5.995 | 0.335 | 0.590 | 11.176 | 0.545 | 0.496 |

Table 1: **Evaluation of the proposed regularizer over the iPhone dataset.** The proposed regularizer improves the performance of all the baselines across all scenes. We were not able to converge DNeRF on some sequences.

| Method | Chickchicken | | Torchocolate | | Aleks-teapot | |
|---|---|---|---|---|---|---|
| | SSIM↑ | LPIPS↓ | SSIM↑ | LPIPS↓ | SSIM↑ | LPIPS↓ |
| TiNeuVox | 0.723 | 0.551 | 0.844 | 0.384 | 0.770 | 0.486 |
| TiNeuVox w/ Reg | 0.796 | 0.381 | 0.859 | 0.377 | 0.834 | 0.441 |

| Method | Lemon | | Hand | | Mean | |
|---|---|---|---|---|---|---|
| | SSIM↑ | LPIPS↓ | SSIM↑ | mLPIPS↓ | SSIM↑ | LPIPS↓ |
| TiNeuVox | 0.816 | 0.662 | 0.718 | 0.667 | 0.774 | 0.550 |
| TiNeuVox w/ Reg | 0.826 | 0.648 | 0.824 | 0.650 | 0.872 | 0.499 |

Table 2: **Evaluation of the proposed regularizer over the HyperNeRF interpolation dataset.**

| Kernel | mPSNR↑ | mSSIM↑ | mLPIPS↓ |
|---|---|---|---|
| Spline | 10.113 | 0.441 | 0.599 |
| Inverse quadratic | 10.441 | 0.423 | 0.501 |
| Gaussian | 11.001 | 0.521 | 0.622 |
| Cubic | 9.993 | 0.425 | 0.513 |
| Quintic | 9.336 | 0.415 | 0.581 |
| Multiquadric | 10.661 | 0.445 | 0.522 |
| Linear | 11.176 | 0.545 | 0.496 |

Table 3: **Comparison over different interpolation kernels.** A linear kernel performed best.

| Regularization | mPSNR↑ | mSSIM↑ | mLPIPS↓ |
|---|---|---|---|
| TiNeuVox (baseline) | 9.444 | 0.504 | 0.536 |
| Scene-flow | 9.457 | 0.514 | 0.536 |
| LPIPS | 9.991 | 0.521 | 0.512 |
| Depth | 9.437 | 0.519 | 0.518 |
| RB | 9.332 | 0.518 | 0.518 |
| SR | 10.131 | 0.523 | 0.512 |
| RB + SR + Depth | 10.311 | 0.535 | 0.508 |
| Ours | 11.176 | 0.545 | 0.496 |
| RB + SR + Depth + Ours | 11.221 | 0.565 | 0.490 |

Table 4: **Comparison over different regularizations.** Depth and Scene-flow regularizations include expensive preprocessing steps that involve evaluating off-the-shelf models. LPIPS regularization requires stacking a pre-trained deep network on NeRF at training time. In contrast, our regularization is more efficient and performs better.

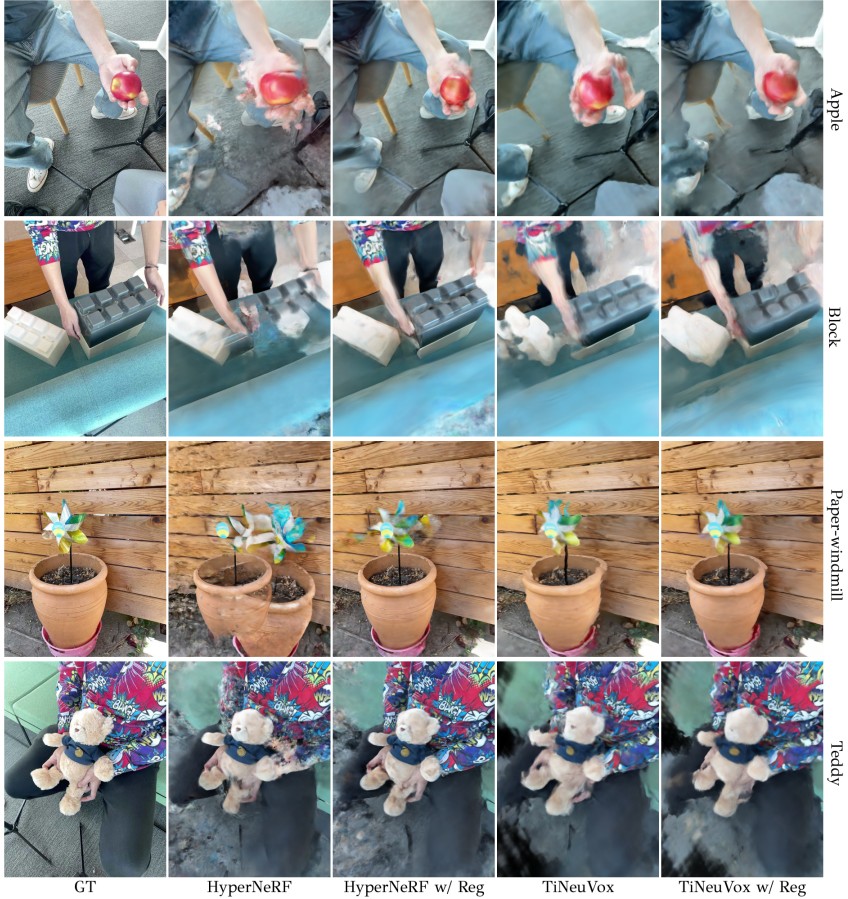

Figure 3: **A qualitative illustration of the effect of the proposed regularization.** As illustrated, our regularizer significantly stabilizes the 3D reconstruction task.

uation setting, we measure perceptual similarity between the ground truth image at time zero and the rendered images at different time stamps where the camera pose is fixed at the first ground truth pose. The quantitative and qualitative results are shown in Table 2 and Fig. 4, respectively. Interestingly, we observed that HyperNeRF and Nerfies baselines already performed well in this setting and our regularization did not have much effect. In contrast, TiNeuVox demonstrated difficulty in converging to a plausible solution and our regularizer significantly improved the results.

### 4.4 ABLATION AGAINST OTHER REGULARIZATIONS AND DESIGN CHOICES

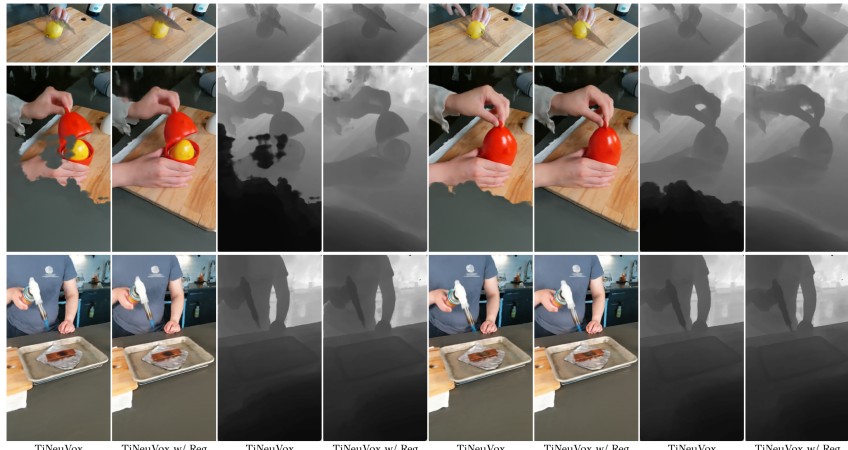

| TiNeuVox | TiNeuVox w/ Reg | TiNeuVox | TiNeuVox w/ Reg | TiNeuVox | TiNeuVox w/ Reg | TiNeuVox | TiNeuVox w/ Reg |

Figure 4: A qualitative illustration of the effect of the proposed regularization on the HyperNeRF interpolation dataset.

We compare the efficacy of the proposed method against other regularizations. Results are shown in Table 4. For depth regularization, we use the LIDAR-based depth maps provided by the iPhone dataset for computing the photometric error between the rendered depth and ground-truth depths. For LPIPS regularization, we minimize the LPIPS distance between RGB patches rendered from a fixed camera, in different time stamps (similar to our regularization, but LPIPS is used instead of the OT distance). We used a pre-trained AlexNet for computing LPIPS. For scene-flow, we use the method proposed in Wang et al. (2023) which uses RAFT (Teed & Deng, 2020) for estimating optical flow. We also compared against a

| Method | mPSNR↑ | mSSIM↑ | mLPIPS↓ |
|---|---|---|---|
| Baseline | 9.444 | 0.504 | 0.536 |
| $L2$ | 8.313 | 0.407 | 0.697 |
| KL-divergence | 9.414 | 0.461 | 0.570 |
| Total variation norm | 9.471 | 0.426 | 0.597 |
| Gaussian-MMD | 10.011 | 0.523 | 0.511 |
| Laplacian-MMD | 10.114 | 0.522 | 0.500 |
| Energy-MMD | 10.746 | 0.513 | 0.521 |
| Sinkhorn | 10.888 | 0.533 | 0.498 |
| Sliced-Wasserstein | 11.176 | 0.545 | 0.496 |

Table 5: **Comparison over different distant metrics used with the proposed regularization.** Sliced-Wasserstein yields the best results.

sparcity regularizer Barron et al. (2022) and random background regularizer Weng et al. (2022) which has been previously been used in the literature. Our regularization outperforms all the above methods that use either additional devices, off-the-shelf models, or deep networks. Table 5 and Table 3 depict comparison against other distance metrics and interpolation kernels. Interestingly, sliced-Wasserstein outperformed all other metrics by a significant margin. We use TiNeuVox on the iPhone dataset for these experiments.

## 5 LIMITATIONS

Our method assumes smooth motions and can fail with abrupt scene dynamics. However, note that in most real world scenes, this is a fair assumption. Additionally, due to our method's reliance on an approximate OT distance, a pixel averaging effect can occasionally occur, resulting in a slight blur. Interestingly, this effect can potentially lead to a minor degradation in renderings, particularly when the baseline model already performs well in a given sequence. Moreover, if the baseline model fails to converge entirely in a specific sequence, our regularization method may not yield significant improvements in the results.

## 6 CONCLUSION

We propose an architecture-agnostic, simple regularizer that can be easily integrated into dynamic NeRF models. We do not rely on deep architectures or expensive learned priors across a large dataset. We leverage optimal transport (OT) to learn instance-based statistical priors on-the-fly during training. To circumvent the need to solve an OT problem in each iteration, we employ a sliced-Wasserstein approximation and derive theoretical bounds for to the convergence with respect to sampling complexity. We show the effectiveness of our regularizer by evaluating it across challenging real-world dynamic scenes.

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

# A  APPENDIX

## A.1  PROOF OF THEOREM 1

In order to prove our result, we introduce some additional notation. For any probability measure $\mu \in \mathcal{P}(\mathbb{R})$, the *support* of the measure $\mu$ is

$$\text{supp}(\mu) := \{\mathsf{X} \in \mathbb{R} : \text{if } \mathsf{X} \in N_{\mathsf{X}} \text{ open} \implies \mathcal{P}(N_{\mathsf{X}}) > 0\}. \tag{4}$$

Furthermore, for any set $A \subseteq \mathbb{R}$, we use $1_A(x) = 1$ if $x \in A$ and 0 otherwise to denote the characteristic function of the set $A$. Furthermore, we use $[\![\cdot]\!]$ to denote the Iverson bracket of an event: $[\![A]\!] = 1$ if $A$ is true and 0 otherwise. We will require the use of a Lemma that will aid us in the main proof

**Lemma 1** *For any measure $\mu \in \mathcal{P}([0,1))$ with finite support: $|\text{supp}(\mu)| < \infty$, let $\hat{\mu}_n = \frac{1}{n}\sum_{i=1}^{n} \delta_{x_i}$ where $x_i \sim \mu$ i.i.d, then we have*

$$|\mu(x) - \hat{\mu}_n(x)| \leq \sqrt{\frac{1}{2n}\log\left(\frac{2}{\delta}\right)}, \tag{5}$$

*for a fixed $x \in \mathbb{R}$ with probability $1 - \delta$.*

**Proof** Note that $\mu(x) = \mathbb{E}[\hat{\mu}_n(x)]$ and $\hat{\mu}_n(x) \in [0,1]$. Thus by a standard concentration inequality such as Hoeffding's inequality, we get

$$\mathbb{P}\left[|\hat{\mu}_n(x) - \mu(x)| \geq t\right] \leq 2\exp\left(-2nt^2\right). \tag{6}$$

Setting $t = \sqrt{\frac{1}{2n}\log(2/\delta)}$ completes the proof. ∎

**Lemma 2** *For any measure $\mu \in \mathcal{P}([0,1))$ with finite support: $|\text{supp}(\mu)| < \infty$, let $\hat{\mu}_n = \frac{1}{n}\sum_{i=1}^{n} \delta_{x_i}$ where $x_i \sim \mu$ i.i.d, then we have*

$$W_1(\mu, \hat{\mu}_n) \leq \sqrt{\frac{|\text{supp}(\mu)|^2}{2n}\log\left(\frac{2 \cdot |\text{supp}(\mu)|}{\delta}\right)}, \tag{7}$$

*with probability at least $1 - \delta$.*

**Proof** We first invoke the Kantorovich-Rubenstein dual of the Wasserstein distance (Villani et al., 2009):

$$W_1(\mu, \hat{\mu}_n) = \sup_{h:[0,1]\to\mathbb{R}:|h(x)-h(x')|\leq|x-x'|} (\mathbb{E}_\mu[h] - \mathbb{E}_{\hat{\mu}_n}[h]) \tag{8}$$

$$\overset{(1)}{\leq} \sup_{h:[0,1]\to\mathbb{R}:|h(x)|\leq 1} (\mathbb{E}_\mu[h] - \mathbb{E}_{\hat{\mu}_n}[h]) \tag{9}$$

$$\overset{(2)}{=} \sup_{A\subseteq[0,1]} (\mathbb{E}_\mu[\mathbf{1}_A] - \mathbb{E}_{\hat{\mu}_n}[\mathbf{1}_A]) \tag{10}$$

$$\overset{(3)}{=} \sup_{A\subseteq[0,1]} \left( \sum_{x\in\text{supp}(\mu)} [\mu(x) \cdot \mathbf{1}_A(x)] - \sum_{x\in\text{supp}(\mu)} [\hat{\mu}_n(x) \cdot \mathbf{1}_A(x)] \right) \tag{11}$$

$$= \sup_{A\subseteq[0,1]} \left( \sum_{x\in\text{supp}(\mu)} [\mu(x) - \hat{\mu}_n(x)] \cdot \mathbf{1}_A(x) \right), \tag{12}$$

where (1) is due to fact that $|x - x'| \leq 1$ for $x, x' \in [0,1]$, (2) is by dual formulation of the Total Variation, and (3) is by the fact that $\hat{\mu}_n$ is support on $\text{supp}(\mu)$ by construction. Consider then the following boolean variable

$$\mathsf{H}_x = [\![|\mu(x) - \hat{\mu}_n(x)| > \sqrt{\frac{1}{2n}\log\left(\frac{2 \cdot |\text{supp}(\mu)|}{\delta}\right)}]\!], \tag{13}$$

we then have by Union bound inequality of probability:

$$\mathbb{P}\left[\bigcup_{x\in\text{supp}(\mu)}\mathsf{H}_x\right] \leq \sum_{x\in\text{supp}(\mu)}\mathbb{P}\left[\mathsf{H}_x\right] \tag{14}$$

$$\leq \sum_{x\in\text{supp}(\mu)}\frac{\delta}{|\text{supp}(\mu)|} \tag{15}$$

$$= \delta. \tag{16}$$

Therefore, we have

$$\mathbb{P}\left[W_1(\mu,\hat{\mu}_n) \geq \sqrt{\frac{|\text{supp}(\mu)|^2}{2n}\log\left(\frac{2\cdot|\text{supp}(\mu)|}{\delta}\right)}\right] \tag{17}$$

$$\geq \mathbb{P}\left[\sup_{A\subseteq[0,1]}\left(\sum_{x\in\text{supp}(\mu)}[\mu(x)-\hat{\mu}_n(x)]\cdot\mathbf{1}_A(x)\right) \geq \sqrt{\frac{|\text{supp}(\mu)|^2}{2n}\log\left(\frac{2\cdot|\text{supp}(\mu)|}{\delta}\right)}\right] \tag{18}$$

$$\overset{(1)}{\geq} \mathbb{P}\left[\bigcup_{x\in\text{supp}(\mu)}\mathsf{H}_x\right] \tag{19}$$

$$\geq \delta, \tag{20}$$

where $(1)$ is due to the union bound argument, completing the proof. ∎

We are now ready to complete the proof:

$$W_1(\mathcal{P}(\{\hat{\mathbf{r}}^{t_1}\}),\mathcal{P}(\{\hat{\mathbf{r}}^{t_2}\})) \leq W_1(\mathcal{P}(\{\hat{\mathbf{r}}^{t_1}\}),\mathcal{P}(\{\mathbf{r}^{t_1}\})) + W_1(\mathcal{P}(\{\mathbf{r}^{t_1}\}),\mathcal{P}(\{\hat{\mathbf{r}}^{t_2}\})) \tag{21}$$

$$\leq W_1(\mathcal{P}(\{\hat{\mathbf{r}}^{t_1}\}),\mathcal{P}(\{\mathbf{r}^{t_1}\})) \tag{22}$$

$$+ W_1(\mathcal{P}(\{\hat{\mathbf{r}}^{t_2}\}),\mathcal{P}(\{\mathbf{r}^{t_2}\})) + W_1(\mathcal{P}(\{\mathbf{r}^{t_1}\}),\mathcal{P}(\{\mathbf{r}^{t_2}\})) \tag{23}$$

$$\leq 4\sqrt{\frac{|\text{supp}(\mu)|^2}{2n}\log\left(\frac{2\cdot|\text{supp}(\mu)|}{\delta}\right)} + W_1(\mathcal{P}(\{\mathbf{r}^{t_1}\}),\mathcal{P}(\{\mathbf{r}^{t_2}\})). \tag{24}$$

Similarly, we can apply the above argument to get

$$W_1(\mathcal{P}(\{\mathbf{r}^{t_1}\}),\mathcal{P}(\{\mathbf{r}^{t_2}\}) \leq 4\sqrt{\frac{|\text{supp}(\mu)|^2}{2n}\log\left(\frac{2\cdot|\text{supp}(\mu)|}{\delta}\right)} + W_1(\mathcal{P}(\{\hat{\mathbf{r}}^{t_1}\}),\mathcal{P}(\{\hat{\mathbf{r}}^{t_2}\})). \tag{25}$$

Merging these two together allows us to complete the proof.

## A.2 EVALUATION ON HYPERNERF VRIG DATASET

The HyperNeRF dataset, initially introduced by Park et al. (2021a) and Park et al. (2021b), underwent revisions after Gao et al. (2022) identified certain limitations. These limitations included frames that transitioned abruptly between multiple camera viewpoints in consecutive time steps, a scenario challenging to capture from a single camera, as well as scenes portraying quasi-static scenarios that do not accurately represent real-world dynamics. In response, Gao et al. (2022) proposed an enhanced and more demanding version of this dataset, which we employ for our evaluation. This augmented dataset comprises seven sequences in total, each enriched with keypoint annotations. It encompasses 7 multi-camera captures and 7 single-camera captures, all featuring 480p resolution videos. It is noteworthy that all dynamic scenes within this dataset are inward-facing. For our evaluation, we apply masked metrics as introduced in Gao et al. (2022), which utilize covisibility masks. Results are shown in Table 6 and Fig. 5.

| Method | Broom | | | Chicken | | | Peel-banana | | |
|---|---|---|---|---|---|---|---|---|---|
| | mPSNR↑ | mSSIM↑ | mLPIPS↓ | mPSNR↑ | mSSIM↑ | LPIPS↓ | mPSNR↑ | mSSIM↑ | mLPIPS↓ |
| TiNeuVox | 19.633 | 0.609 | 0.659 | 24.103 | 0.777 | 0.264 | 21.558 | 0.863 | 0.267 |
| TiNeuVox w/ Reg | 20.955 | 0.774 | 0.628 | 26.006 | 0.949 | 0.196 | 22.313 | 0.891 | 0.274 |

| Method | 3dprinter | | | Tail | | | Toby sit | | |
|---|---|---|---|---|---|---|---|---|---|
| | mPSNR↑ | mSSIM↑ | mLPIPS↓ | mPSNR↑ | mSSIM↑ | mLPIPS↓ | mPSNR↑ | mSSIM↑ | mLPIPS↓ |
| TiNeuVox | 19.184 | 0.780 | 0.296 | 22.593 | 0.822 | 0.511 | 18.191 | 0.783 | 0.611 |
| TiNeuVox w/ Reg | 19.323 | 0.775 | 0.318 | 23.191 | 0.891 | 0.438 | 19.385 | 0.790 | 0.536 |

Table 6: **Evaluation of the proposed regularizer over the HyperNeRF vrig dataset.**

| Method | $d = 128$ | | | $d = 256$ | | | $d = 512$ | | |
|---|---|---|---|---|---|---|---|---|---|
| | mPSNR↑ | mSSIM↑ | mLPIPS↓ | mPSNR↑ | mSSIM↑ | LPIPS↓ | mPSNR↑ | mSSIM↑ | mLPIPS↓ |
| TiNeuVox w/ Reg | 9.441 | 0.409 | 0.575 | 10.020 | 0.478 | 0.510 | 10.190 | 0.479 | 0.536 |

| Method | $d = 1024$ | | | $d = 2048$ | | | $d = 4096$ | | |
|---|---|---|---|---|---|---|---|---|---|
| | mPSNR↑ | mSSIM↑ | mLPIPS↓ | mPSNR↑ | mSSIM↑ | mLPIPS↓ | mPSNR↑ | mSSIM↑ | mLPIPS↓ |
| TiNeuVox w/ Reg | 10.325 | 0.489 | 0.544 | 10.841 | 0.525 | 0.555 | 11.176 | 0.545 | 0.496 |

Table 7: **Ablation over the number of samples used for OT distance computation.** Metrics are calculated over the iPhone dataset. As the number of samples grow, the performance improves, possible due to the lower error in OT computation as predicted by Theorem 1.

## A.3 EFFICIENCY OF OUR REGULARIZER

The efficiency of our regularization approach is attributed to two key factors: 1) the elimination of data pre-processing and 2) superior run-time computational efficiency.

For depth regularization, the main efficiency bottleneck is the necessity to pre-compute depth maps for all frames. Although this adds minimal computational overhead during run-time, it is a significant pre-processing step. In contrast, our method requires no such pre-processing, offering a clear advantage.

More importantly, our method demonstrates substantial training time efficiency compared to both scene-flow and LPIPS regularizations. The following table illustrates the average run-time for the TiNeuVox on the iPhone dataset:

| Method | Time Required |
|---|---|
| Ours | $\sim$ 3 hours |
| LPIPS | $\sim$ 8 hours |
| Scene-flow | $\sim$ 10 hours |

Table 8: Comparison of Time Required for Different Methods

Scene-flow regularization demands pre-computing optical flows and solving integration problems using the Runge-Kutta method during training, which is computationally intensive. Similarly, LPIPS involves backpropagating through an additional deep network at training, adding to its computational load.

In contrast, our method is significantly more lightweight, as it avoids these computationally demanding steps.

## A.4 FAILURE CASES

Since our method relies on an approximate OT distance, there can be instances where a pixel averaging effect occurs, resulting in a subtle blur. Interestingly, this effect can potentially cause a slight decline in the quality of renderings, especially when the baseline model already performs exceptionally well in a given sequence. Additionally, if the baseline model fails to converge completely in a particular sequence, our regularization technique may not yield substantial improvements in the results. Fig. 7 shows some examples.

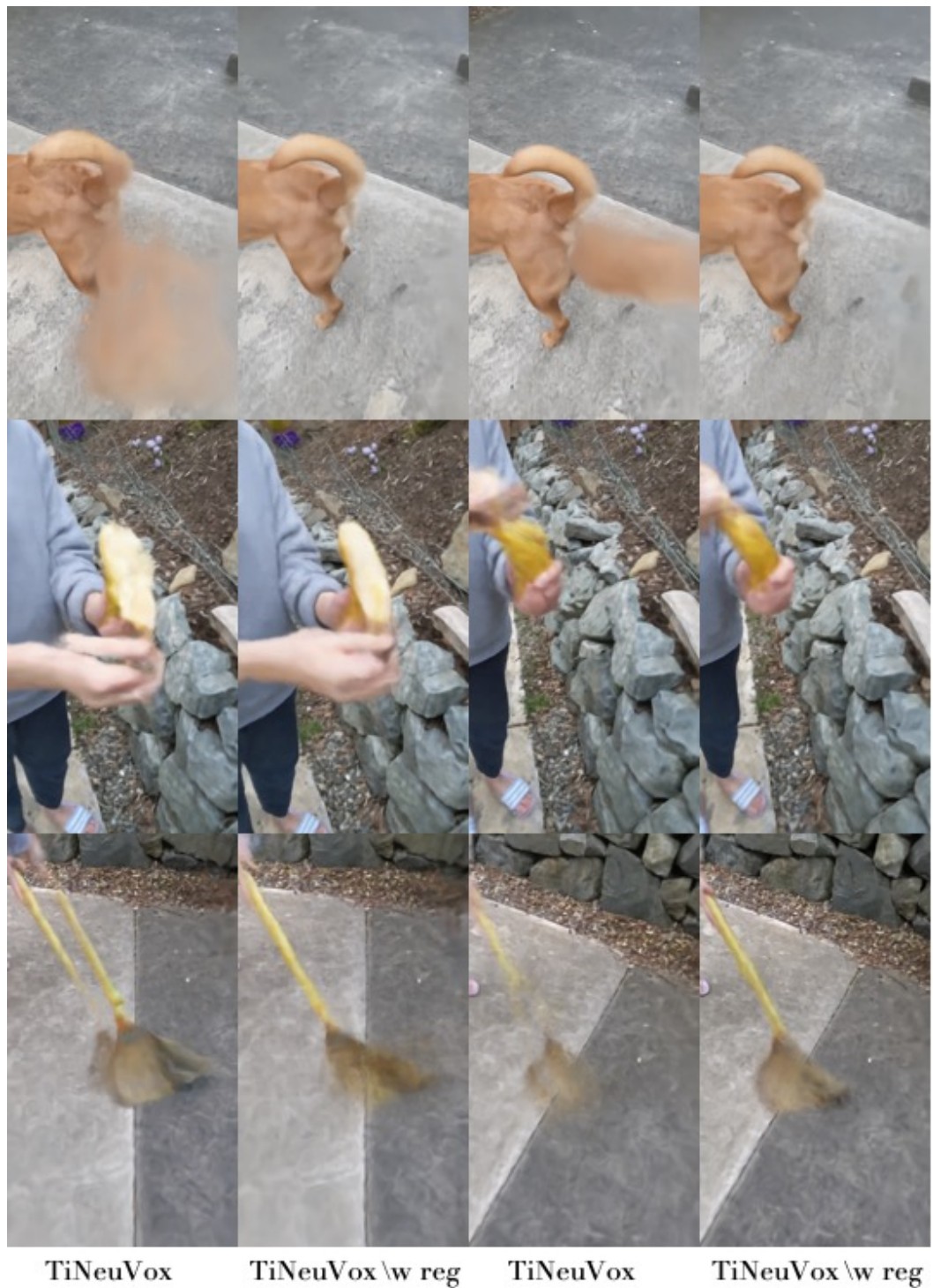

Figure 5: **A qualitative illustration of the effect of the proposed regularization on the HyperNeRF vrig dataset.**

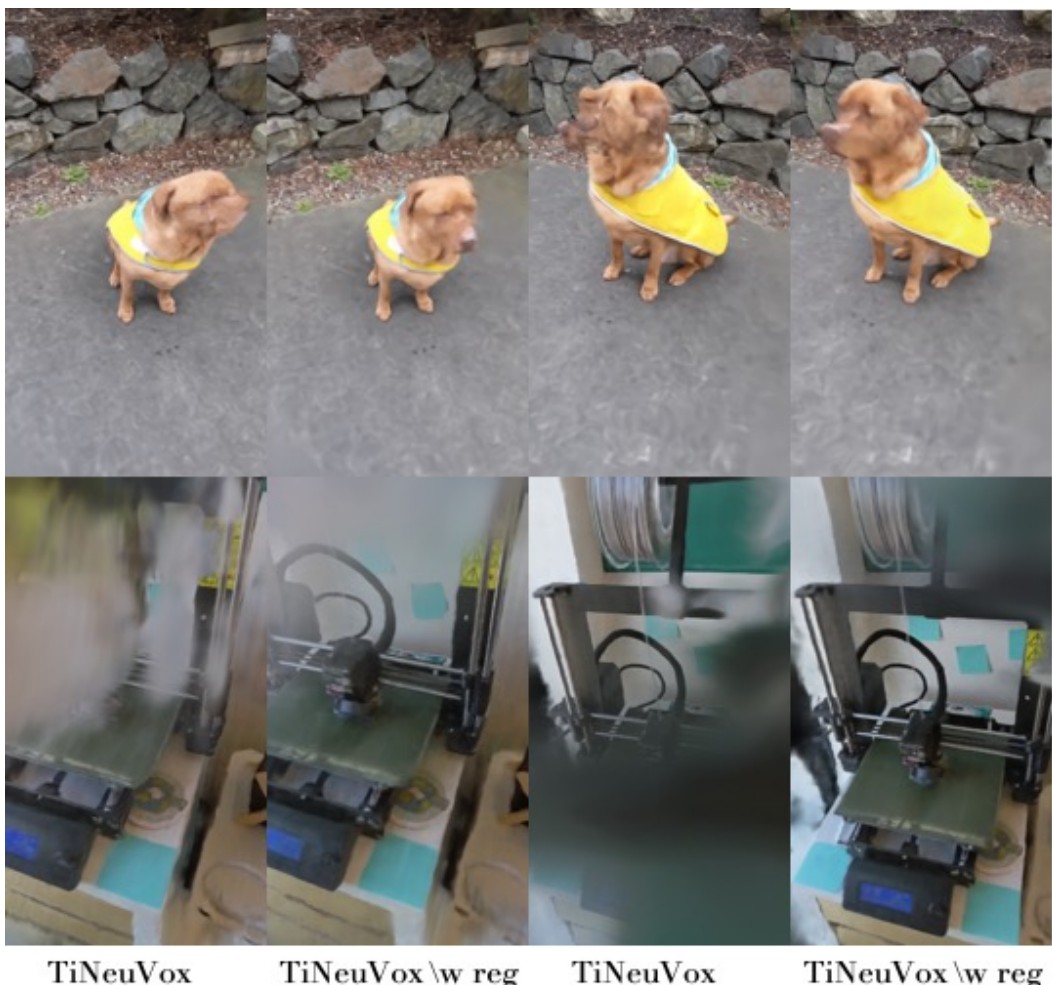

TiNeuVox     TiNeuVox \w reg     TiNeuVox     TiNeuVox \w reg

Figure 6: **A qualitative illustration of the effect of the proposed regularization on the HyperNeRF vrig dataset.**

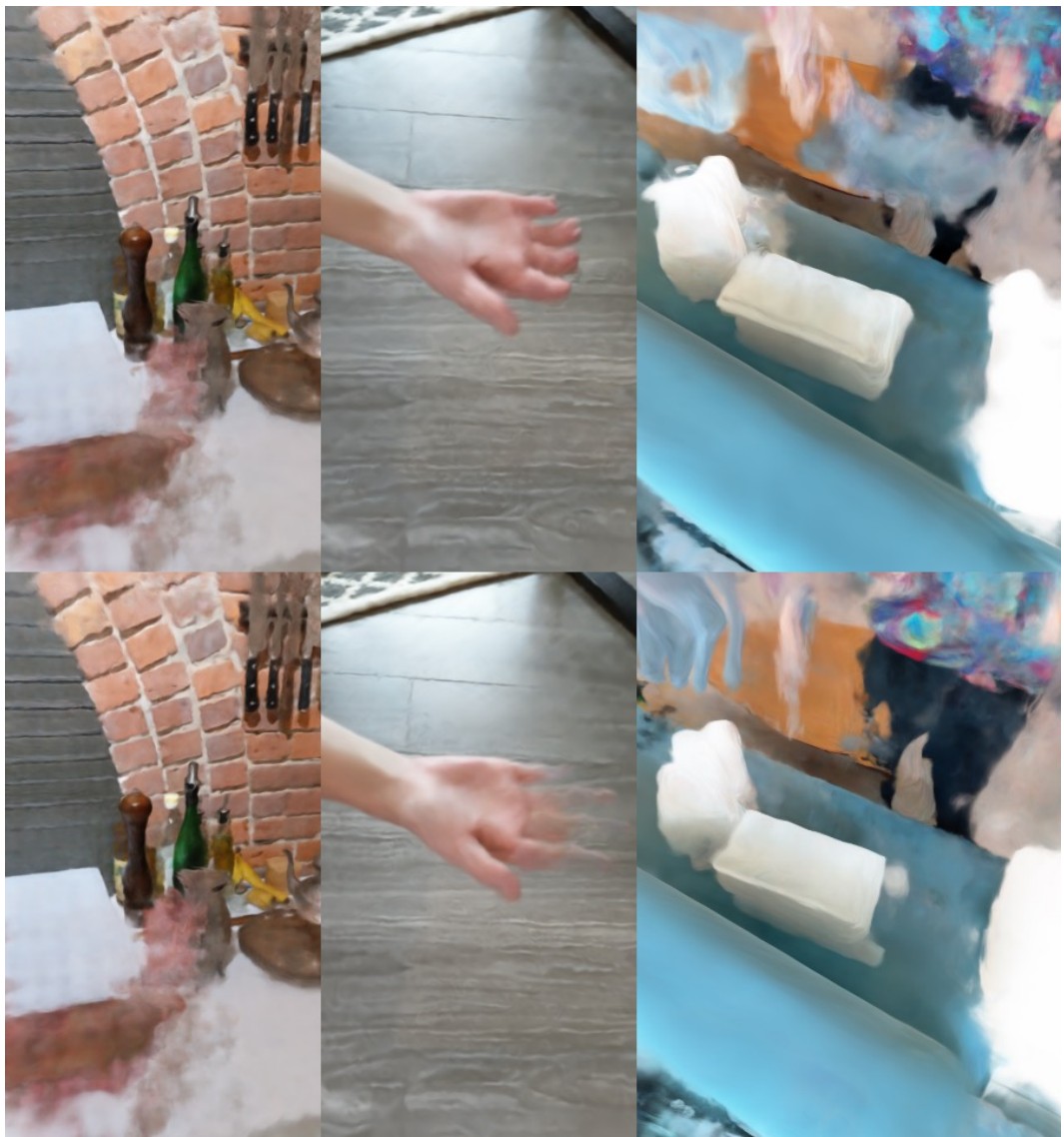

Figure 7: **Example failure cases.** *Top row*: baseline, *bottom row*: baseline with regularization. Due to the pixel averaging effect that stems from the OT approximation sometimes degrade or does not improve the results. We observed cases where the baseline model performs too poorly, leading to our regularization not having much effect.

