# OpenReview forum: "Improving the Convergence of Dynamic NeRFs via Optimal Transport"
_ICLR.cc/2024/Conference — ICLR 2024 poster_

### Official Review · Reviewer_bXej · 2023-10-25

**Soundness:** 3 good
**Presentation:** 3 good
**Contribution:** 3 good
**Rating:** 6
**Confidence:** 2

**Summary:**

This paper proposed a regularization method for dynamic NeRFs using the optimal transport of pixel values at two different time steps, exploiting consistent structures between the video frames at different time steps. Although they assumed smooth motions without abrupt scene dynamics, they showed consistent results for iPhone and HyperNeRF interpolation datasets applying to various dynamic NeRFs.

**Strengths:**

- A simple yet effective regularization for dynamic NeRFs.

- Thank you for the neat presentation of your work in the code of Alg. 1.

**Weaknesses:**

- As the authors mentioned, this method asserted smooth motions without abrupt scene dynamics in the dynamic scene. The proposed method is rather an ad-hoc method for a subset of dynamic scenes. The authors may introduce a method to determine whether a given scene is appropriate to apply this method before full training and evaluation.

**Questions:**

- Q1. In Tbl. 1. Hexplane w/ Reg on the Block scene had a significant gain while deteriorating the other scenes (i.e., Paper windmill, Teddy, Wheel in PSNR). How do you speculate on this result?

- Q2. In Sec. 3.1, the theoretical statement makes the paper unnecessarily complicated. And I do not follow why Thm. 1 is necessary for the argumental context.

- Q3. In Fig. 1, how do you decide which metric is the best in the manner of quantitativeness?

- Minors:
  - In Sec. 2, Related works -> Related work
  - In Sec. 3.1, from an images -> from images
  - The caption of Tbl. 5 goes beyond the content boundary of the bottom.

---

> ### Author Response · Authors · 2023-11-15
> **Response to reviewer bXej**
>
> We thank the reviewer for the encouraging and valuable comments. Please find our answers below.
>
> **The authors may introduce a method to determine whether a given scene is appropriate to apply this method before full training and evaluation.**
>
> Thank you for the interesting suggestion. Please note that predicting the regularizer's effectiveness a priori necessitates quantifying object deformation speeds from monocular images, which is challenging. This task is notably more complex than the motion magnitude measurement of the camera as which is attempted by Dycheck [1]. A tentative approach, however, could involve identifying 2D point correspondences across images, lifting them to 3D using an off-the-shelf lifting method [2, 3], and then quantifying deformation using these sparse points. This approach, while promising, is complex and extends beyond the scope of our current work. We believe it presents an exciting avenue for future research.
>
> On the other hand, we do acknowledge that this is a limitation of the hypothesis is smooth motion. However, note that this limitation is not unique to our regularizer. Other dynamic NeRFs (and the regularizers proposed therein) themselves work under the assumption that the motion is smooth. For example, scene-flow regularizers diverge under large movement (NR-NeRF) and depth regularizers (Dycheck) will fail due to fast movement which result in occlusions and signal mismatch. For all these cases, the deformation of light and density fields is difficult to be modeled by neural networks which have finite bandwidth. Therefore, this is a standard and a fair assumption in modeling most real world 3D scenes.
>
> [1] - Gao, Hang, et al. "Monocular dynamic view synthesis: A reality check." Advances in Neural Information Processing Systems 35 (2022): 33768-33780.
>
> [2] - Nie, Qiang, Ziwei Liu, and Yunhui Liu. "Lifting 2D Human Pose to 3D with Domain Adapted 3D Body Concept." International Journal of Computer Vision 131.5 (2023): 1250-1268.
>
> [3] - Llopart, Adrian. "LiftFormer: 3D Human Pose Estimation using attention models." arXiv e-prints (2020): arXiv-2009.
>
> **Q1. In Tbl. 1. Hexplane w/ Reg on the Block scene had a significant gain while deteriorating the other scenes (i.e., Paper windmill, Teddy, Wheel in PSNR). How do you speculate on this result?**
>
> We agree with the reviewer that this needs further investigation.  Hexplanes performed poorly on some sequences. It has been also observed by others that Hexplanes may not work well in monocular settings : \url{https://github.com/Caoang327/HexPlane/issues/7#issuecomment-1616084851}. This maybe due to a model bias of Hexplanes that may be favorable to some scenes.
>
> **Q2. In Sec. 3.1, the theoretical statement makes the paper unnecessarily complicated. And I do not follow why Thm. 1 is necessary for the argumental context.**
>
> We appreciate your feedback on Section 3.1 and Theorem 1. We would like to clairfy that Theorem 1 provides an essential validity for our paper. The essence of Theorem 1 lies in its demonstration that **random sampling** from pixel distributions yields an approximate measure of the Optimal Transport (OT) distance. This is the fundamental aspect distinguishing our method from deep learning-based pixel statistic losses, such as LPIPS. Unlike LPIPS, which requires  densely sampled patches --- which only reflects local pixel statistics --- our approach captures the global statistics of the frame more efficiently (e.g., calculating LPIPS similarity between randomly sampled pixels from images
> lacks meaning). For LPIPS to capture global statistics, the entire frame  has to be rendered at each trainin step, which is computationally infeasible.
> While we agree with the reviewer that this result might be intuitive, we also believe its formal, rigorous proof adds scientific value to our work.
>
> Further, this particular convergence bound in the context of finitely supported distributions has not been previously shown in the literature, to our knowledge. We demonstrate a convergence rate of $O(1/\sqrt{n})$, which is rapid.  Additionally, we are the first to have only a polylogarithmic dependence on the size of support of the distribution $S$, namely $O(S \sqrt{\log S})$. This finding is not just a theoretical advancement but also has practical implications for optimizing model performance in realistic settings which we believe would provide grounding for future works as well. In a practical point of view, Table 7 demonstrates how our method's overall ability to regularize dynamics increases with the sample size, as indicated by the theory.

---

> ### Author Response · Authors · 2023-11-16
> **Response to reviewer bXej Part 2**
>
> **In Fig. 1, how do you decide which metric is the best in the manner of quantitativeness?**
>
> It is important to note that direct comparison across different distance metrics is not feasible due to their unique scales. Instead, the intention of Figure 1 is to encourage the readers (as a toy example) to assess their performance through visual observation of heatmap distributions. The primary aim of Figure 1 is to intuitively demonstrate that geometric distances are more effective for this specific problem than pixel-to-pixel distances, such as the $L2$ norm. Additionally, it visually indicates that the sliced-Wasserstein distance slightly outperforms others. This particular advantage of the sliced-Wasserstein distance over alternative metrics is further empirically quantified in Table 5.
>
> **Minors**
>
> Thank you for carefully going through our paper. We have addressed all the mentioned issues.

---

> > ### Comment · Reviewer_bXej · 2023-11-20
> > **The rebuttal to be considered**
> >
> > Thanks for the author's feedback, and no more inquiries from me. It will be considered in the reviewer discussion period.

---

### Official Review · Reviewer_HrLW · 2023-10-30

**Soundness:** 4 excellent
**Presentation:** 4 excellent
**Contribution:** 2 fair
**Rating:** 6
**Confidence:** 4

**Summary:**

This paper introduces a regularization method for dynamic view synthesis by reformulating the regularization task to an optimal transportation problem. The paper assumes that, for a fixed camera pose, the intensity of the pixel of images within a small interval should be constant. Hence, the authors propose to minimize the divergence metric of two distributions at two times. The method is architecture-agnostic, simple and can be easily integrated into existing frameworks. The experiments show that the proposed regularization improve the performance of different frameworks.

**Strengths:**

1.	The work reformulates the problem of regularization of dynamic nerf based on the assumption proposed by the paper. The Fig 1 clearly shows the difference between the proposed method and the simple metrics (like L2 distance).
2.	The proposed method seems technically sound with the theory of optimal transportation.
3.	The unbiased estimator is a clear advantage over existing regularization, like depth and optical flow.

**Weaknesses:**

1.	I am not the expert of the optimal transportation area. I cannot judge the novelty of the designed method, including the proposed theorem 1, and the sliced-Wasserstein approximator. It seems that the optimal transport method proposed here is not original and lack novelty. But again, I am not an expert for this area, and I would be open mind for other reviewers’ opinions for this part.
2.	The proposed regularizations only based on 2D distributions and cannot regularize 3D deformation directly. For me, it is equal to the idea that keep the image as static as possible. Is it true that the improvement of the regularization come from the improvement of the static background of the images, which are not related to the dynamic objects? The chichchicken video in supp supports this point.
3.	This method does not have much effect when the performance of the framework is above the certain level.
4.	This method could not deal with scenes with high frequency details, as such situation could break the assumption of the paper.
5.	There should be more visual results, especially for videos. Combine the two videos side by side in one video would be much helpful for comparison.

**Questions:**

1.	What are the settings of the baseline methods? Do you remove all other losses and only use the photometric loss, or you use all the proposed losses of the original papers.

---

> ### Author Response · Authors · 2023-11-15
> **Response to reviewer HrLW**
>
> We thank the reviewer for the insightful comments. Please find our answers below.
>
> **I am not the expert of the optimal transportation area. I cannot judge the novelty of the designed method, including the proposed theorem 1, and the sliced-Wasserstein approximator. It seems that the optimal transport method proposed here is not original and lack novelty. But again, I am not an expert for this area, and I would be open mind for other reviewers’ opinions for this part.**
>
> We would like to point out that  we are the first to use optimal transport for improving the convergence of dynamic nerfs, as also acknowledged by Reviewer **cmoJ**.
>
> Our theoretical result is the first to analyze convergence of optimal transport distances in the setting of finitely supported distributions. This has not been shown in the literature before, to the best of our knowledge. In particular, we show a competitive rate of $O(n^{-1/2})$. Additionally, we are the first to have only a polylogarithmic dependence on the size of support of the distribution $S$, namely $O(S \sqrt{\log S})$.
>
> This result is important for our method as it provides the foundation that randomly sampled pixels can approximate the pixel distributions over the entire frame which is not possible when using other pixel statistics based  regularization methods such as LPIPS; LPIPS needs dense pixel patches which does not capture the scene dynamics across an entire frame.  For LPIPS to capture global statistics, the entire frame  has to be rendered at each trainin step, which is computationally infeasible.
>
> **The proposed regularizations only based on 2D distributions and cannot regularize 3D deformation directly. For me, it is equal to the idea that keep the image as static as possible. Is it true that the improvement of the regularization come from the improvement of the static background of the images, which are not related to the dynamic objects? The chichchicken video in supp supports this point.**
>
> We would like to clarify that our method's improvement through regularization is primarily attributed to its ability to accurately capture the **dynamics of a scene**, rather than just the static background elements. This is a key advantage of employing Optimal Transport (OT) measures over traditional pixel-to-pixel measures like the $L2$ loss.
>
> If the method is to try to keep the image "as static as possible", then $L2$ loss should perform better or on par with our method as $L2$ would enforce the rendered pixels to be static over time.
>
> Howeve, as detailed in Table 5, using $L2$ loss for regularization significantly hampers performance because it fails to adequately represent dynamic content within the scene. This limitation is visually demonstrated in Figure 1. Furthermore, our approach reveals a direct correlation between sample size and the model’s proficiency in capturing dynamic content. In scenarios with rapid pixel changes, a smaller sampling size might inadvertently average these changes, leading to a blurring effect on moving objects.
>
> Conversely, as shown in Table 7, increasing the sample size enhances the model’s capability in reconstructing dynamic scenes more accurately. This phenomenon is not just an empirical observation but is also theoretically underpinned by our Theorem 1. Our methodology, therefore, not only addresses the static components of an image but more critically, its dynamic aspects, distinguishing our approach from conventional methods.
>
> **This method could not deal with scenes with high frequency details, as such situation could break the assumption of the paper.**
>
> We acknowledge that this is a limitation of the hypothesis. However, note that this limitation is not unique to our regularizer. Other dynamic NeRFs (and the regularizers proposed therein) themselves work under the assumption that the motion is smooth. For example, scene-flow regularizers diverge under large movement (NR-NeRF) and depth regularizers (Dycheck) will fail due to fast movement which result in occlusions and signal mismatch. For all these cases, the deformation of light and density fields is difficult to be modeled by neural networks which have finite bandwidth. Therefore, this is a standard and a fair assumption in modeling most real world 3D scenes.
>
> An empirical demonstration of this is the improvement our method gets across all the iPhone scenes for all the models. Please note that  iPhone dataset contains natural real-world like scenes. The validity of our assumption is also acknowledged by Reviewer **cmoJ**, where they mention "I think the central hypothesis, i.e., the pixel intensity distribution of a scene should remain approximately consistent within short time intervals, is valid for most cases and existing dynamic scene datasets unless scenes with objects in very high-speeds".

---

> ### Author Response · Authors · 2023-11-15
> **Response to reviewer HrLW - Part 2**
>
> **What are the settings of the baseline methods? Do you remove all other losses and only use the photometric loss, or you use all the proposed losses of the original papers.**
>
> For baselines, we used the exact settings proposed by the original works including all the additional regularization losses. For instance, we used the background regularization loss for HyperNeRF, elastic regularization
> loss for Nerfies, and  TV loss for the TiNeuVox as baselines.

---

### Official Review · Reviewer_C1xG · 2023-11-01

**Soundness:** 2 fair
**Presentation:** 3 good
**Contribution:** 2 fair
**Rating:** 6
**Confidence:** 4

**Summary:**

This paper proposes a regularization method to enhance the quality of novel view synthesis for dynamic NeRFs. The regularization is based on the notion that the light intensity distribution should remain approximately constant from a fixed camera view within short time intervals. Specifically, the method implements distance metrics such as sliced-Wasserstein and serves as a robust plug-in to improve the quality for several state-of-the-art methods across multiple dynamic scene view synthesis benchmarks.

**Strengths:**

- The intuitive idea of exploiting light distribution consistency.
- The proposed regularization is easy to implement and can be integrated into various methods.

**Weaknesses:**

- Baseline comparison. Previous literature, including random background compositing [2], additional metric depth supervision [3] from iPhone sensor, and surface sparsity regularizer [4], presents more advanced regularization techniques. Quantitative results reported in Table 3 and Figure 10 in [1] suggest that the baseline methods Nerfies and HyperNeRF gain a significant performance boost from these techniques. Therefore, it is unclear how well the proposed regularization performs in comparison
- Robustness in combination with other regularizations. It is uncertain whether the proposed regularization term can be effectively combined with those mentioned above [1][2][3] to further enhance the performance.
- Efficiency. The paper lacks data points illustrating the speed and memory usage of the proposed method.

[1] Hang Gao, Ruilong Li, Shubham Tulsiani, Bryan Russell, and Angjoo Kanazawa. Monocular dy-
namic view synthesis: A reality check. *Advances in Neural Information Processing Systems*, 35:
33768–33780, 2022.

[2] Chung-YiWeng,BrianCurless,PratulPSrinivasan,JonathanTBarron,andIraKemelmacher-Shlizerman. Humannerf: Free-viewpoint rendering of moving people from monocular video. In *CVPR*, 2022.

[3]  Wenqi Xian, Jia-Bin Huang, Johannes Kopf, and Changil Kim. Space-time neural irradiance fields for free-viewpoint video. In *CVPR*, 2021.

[4] Jonathan T Barron, Ben Mildenhall, Dor Verbin, Pratul P Srinivasan, and Peter Hedman. Mip-nerf 360: Unbounded anti-aliased neural radiance fields. In *CVPR*, 2022.

**Questions:**

- How does the proposed method compare to more recent regularization techniques?
- Can the proposed method be combined with existing regularization techniques?
- Do we need to compromise performance (reduce batch size to fit into GPU memory) and/or speed (increase per-iteration time) to enable the proposed method?

---

> ### Author Response · Authors · 2023-11-15
> **Response to reviewer C1xG**
>
> We thank the reviewer for the insightful comments. Please find our answers below.
>
> **How does the proposed method compare to more recent regularization techniques?**
>
> We appreciate your suggestion to improve our paper. In our revised paper, we have included a comparison (Table 4) with other regularization methods for TiNeuVox over the iPhone dataset, as detailed below. (RB: random background, SR: Sparcity regularizer).
>
> | Model                             | PSNR   | SSIM  | LPIPS |
> |-----------------------------------|--------|-------|-------|
> | TiNeuVox                          | 15.382 | 0.661 | 0.417 |
> | TiNeuVox, w/depth                 | 15.437 | 0.669 | 0.418 |
> | TiNeuVox, w/RB                    | 15.332 | 0.660 | 0.418 |
> | TiNeuVox, w/SR                    | 16.131 | 0.699 | 0.412 |
> | TiNeuVox, w/RB + SR + depth       | 16.311 | 0.701 | 0.408 |
> | TiNeuVox, w/OT                    | 18.036 | 0.777 | 0.361 |
> | TiNeuVox, w/RB + SR + depth + OT  | 18.101 | 0.778 | 0.360 |
>
> Our findings reveal that 1) our regularization approach performs better than all of the above regularizations combined, and 2) is complementary to existing techniques, effectively improving the results even while working in conjunction with them.
>
> Notably, we would also like to highlight that random background compositing observed to reduce the performance of TiNeuVox, indicating its dependency on specific architectural frameworks. This highlights the universal applicability of our method across different models.
>
> Due to the time constraints of the rebuttal period, we were unable to conduct extensive ablations across other models. However, we acknowledge the importance of this aspect and will include a comprehensive analysis in the camera-ready version of our paper.
>
>
> **Do we need to compromise performance (reduce batch size to fit into GPU memory) and/or speed (increase per-iteration time) to enable the proposed method?**
>
> In our experiments, we adhered to the batch sizes recommended in the original publications for each model. A significant advantage of our model is the minimal memory overhead it introduces. This is primarily because our method does not involve any trainable parameters, thereby eliminating the need for storing or computing additional gradients during the forward pass.
>
> The slight increase in memory usage is due to the storage of 3-dimensional random projection vectors on the GPU, essential for Optimal Transport (OT) computation. However, it's important to note that our method does increase training times. This is attributed to the additional tensor operations required for OT computation. We estimate this increase to be approximately 1.5 times the original training times for each model. However, these training time overheads are minimal compared to other involved regularizations as LPIPS/Scene-flow; average training times for TiNeuVox over iPhone dataset are shown below.
>
> | Method     | Time Required |
> |------------|---------------|
> | Ours       | ~ 3 hours     |
> | LPIPS      | ~ 8 hours     |
> | Scene-flow | ~ 10 hours    |

---

> > ### Comment · Reviewer_C1xG · 2023-11-22
> >
> > I appreciate and acknowledge authors' response, no further questions from my side.

---

### Official Review · Reviewer_cmoJ · 2023-11-01

**Soundness:** 3 good
**Presentation:** 3 good
**Contribution:** 3 good
**Rating:** 6
**Confidence:** 2

**Summary:**

This work presents a regularizer for dynamic NeRFs to improve the rendering quality. Specifically, it is based on the hypothesis that the pixel intensity distribution of a scene, which is rendered from a specific fixed camera view, should remain consistent within short time intervals. Then, this work uses the dissimilarity measure between pixel intensity distributions as the alternative to the pixel-to-pixel distance function to formulate the problem as an Optimal Transport (OT) problem. Additionally, this work employs a sliced-Wasserstein approximation to reduce the OT computation complexity from O(n^3) to O(nlogn), making the proposed regularizer more lightweight as compared to baselines. The experiments on 3 dynamic scene reconstruction datasets valide that the proposed regularizer can further improve SOTA dynamic NeRF models' rendering quality.

**Strengths:**

> + Well-Motivated: I think the central hypothesis, i.e., the pixel intensity distribution of a scene should remain approximately consistent within short time intervals, is valid for most cases and existing dynamic scene datasets unless scenes with objects in very high-speeds. Also, the related works are discussed in an organized way. Specifically, the regularizers in static NeRF as disentangling camera motion from object motion inherently constitute an ill-posed problem, and existing regularizers in dynamic NeRFs with DNN-based solutions are costly and face the domain gap issue. Thus, a lightweight regularize for dynamic NeRF is highly desirable.

> + Interesting problem formulation: I am not familiar with optimal transport (OT) problems, but this work seems to be the first one to formulate the dynamic NeRF regularization as an OT problem and further optimize the corresponding computation complexity in OT to make it lightweight.

> + Solid experiments: the experiments consider 3 dynamic NeRF datasets and 5 dynamic NeRF models, showing impressive improvement across different datasets.

**Weaknesses:**

> + Discussion on Failure Cases: Since the hypothesis of this work is that the pixel intensity distribution of a scene should remain approximately consistent within short time intervals, failure cases may happen when some high-speed objects appear in the scene. However, the limitation section did not add sufficient details about it. It would be better if the author could provide some analysis or experimental results to provide a more direct way to show when the hypothesis will be invalid (e.g., under what speed or in some specific scenes).

> + Comparision with the Strongest Baseline: From Table 1, HexPlane seems to be the strongest baseline, but it was not listed in Tab. 2 and Fig. 3. Could the author add more details about it?

> + Comparison of Efficiency Metrics: The author claimed that their proposed regularization is more efficient. It could be more solid to add the training time to better verify it.

**Questions:**

Minor question: this regularizer seems to be compatible with all dynamic NeRF models because it is defined in the pixel space. Currently, there are more dynamic NeRF models based on Gaussian Spatting (e.g., https://arxiv.org/abs/2308.09713 or https://arxiv.org/abs/2310.08528), will such a new representation cause new challenges in applying the proposed regularize from the optimization side?

---

> ### Author Response · Authors · 2023-11-15
> **Response for the reviewer cmoJ**
>
> We thank the reviewer for the encouraging comments and valuable suggestions. Please find our answers below.
>
> **Discussion on Failure Cases: Since the hypothesis of this work is that the pixel intensity distribution of a scene should remain approximately consistent within short time intervals, failure cases may happen when some high-speed objects appear in the scene. However, the limitation section did not add sufficient details about it. It would be better if the author could provide some analysis or experimental results to provide a more direct way to show when the hypothesis will be invalid (e.g., under what speed or in some specific scenes).**
>
> Thank you for the very interesting suggestion. We have provided some qualitative examples of failure cases in supplementary Figure 7. The failures in the first and last columns (Aleks-teapot and Block) are because the baseline models converged too poorly in those scenes. However, the examples in the middle column (Hand) is illustrates a failure case due to faster deformation of objects.
>
> We further explore the impact of sampling size on our regularizer's efficacy in handling dynamic scenes (refer to Table 7). A larger sampling size enhances the regularizer's ability to better capture movements.
>
> However, predicting the regularizer's effectiveness a priori necessitates quantifying object deformation speeds from monocular images, which is challenging. This task is notably more complex than the motion magnitude measurement of the camera as which is attempted by Dycheck [1]. A tentative approach, however, could involve identifying 2D point correspondences across images, lifting them to 3D using an off-the-shelf lifting method [2, 3], and then quantifying deformation using these sparse points. This approach, while promising, is complex and extends beyond the scope of our current work. We believe it presents an exciting avenue for future research.
>
>
> [1] - Gao, Hang, et al. "Monocular dynamic view synthesis: A reality check." Advances in Neural Information Processing Systems 35 (2022): 33768-33780.
>
> [2] - Nie, Qiang, Ziwei Liu, and Yunhui Liu. "Lifting 2D Human Pose to 3D with Domain Adapted 3D Body Concept." International Journal of Computer Vision 131.5 (2023): 1250-1268.
>
> [3] - Llopart, Adrian. "LiftFormer: 3D Human Pose Estimation using attention models." arXiv e-prints (2020): arXiv-2009.
>
> **Comparision with the Strongest Baseline: From Table 1, HexPlane seems to be the strongest baseline, but it was not listed in Tab. 2 and Fig. 3. Could the author add more details about it?**
>
> The reason we did not include Hexplanes was in Table 2 is that we failed to converge it on the HyperNeRF dataset. This has also been observed by the HexPlane authors themselves in this post:
> \url{https://github.com/Caoang327/HexPlane/issues/7#issuecomment-1616084851}
>  We verified this independently and also tried our regulariser, though the baseline results are too poor for our regulariser to be effective. Our regulariser will not be effective if a baseline model fails to converge on a sequence entirely as we have mentioned in the limitations. We will try our best to add qualitative results for HexPlane during the rebuttal time period.
>
> **Comparison of Efficiency Metrics: The author claimed that their proposed regularization is more efficient. It could be more solid to add the training time to better verify it.**
>
> The efficiency of our regularization approach is attributed to two key factors: 1) the elimination of data pre-processing and 2) superior run-time computational efficiency.
>
> For depth regularization, the main efficiency bottleneck is the necessity to pre-compute depth maps for all frames. Although this adds minimal computational overhead during run-time, it is a significant pre-processing step (if the depth is not obtained during the capture process using a depth sensor). In contrast, our method requires no such pre-processing, offering a clear advantage.
>
> More importantly, our method demonstrates substantial training time efficiency compared to both scene-flow and LPIPS regularizations. The following table illustrates the average run-time for the TiNeuVox on the iPhone dataset:
>
> | Method     | Time Required |
> |------------|---------------|
> | Ours       | ~ 3 hours     |
> | LPIPS      | ~ 8 hours     |
> | Scene-flow | ~ 10 hours    |
>
> Scene-flow regularization demands pre-computing optical flows and solving integration problems using the Runge-Kutta method during training, which is computationally intensive. Similarly, LPIPS involves backpropagating through an additional deep network at training, adding to its computational load.
>
> In contrast, our method is significantly more lightweight, as it avoids these computationally demanding steps. We have included these comparative results in the supplementary material. Thank you for the suggestion.

---

> ### Author Response · Authors · 2023-11-15
> **Response for the reviewer cmoJ - part 2**
>
> **Minor question: this regularizer seems to be compatible with all dynamic NeRF models because it is defined in the pixel space. Currently, there are more dynamic NeRF models based on Gaussian Spatting (e.g., https://arxiv.org/abs/2308.09713 or https://arxiv.org/abs/2310.08528), will such a new representation cause new challenges in applying the proposed regularize from the optimization side?**
>
> Both of these recent works use Gaussian-Splatting formulation to represent dynamic scenes. We would like to point out that our regularizer operates on the final output and is representation agnostic. As such, for both of these works adding our OT regularizer is straight-forward. This indicates a key advantage of our method against more intricate regularizations that use, for instance, scene-flows/optical flows.

---

### Meta-Review · Area_Chair_hyUb · 2023-12-08

**Metareview:**

This paper examines the task of learning (neural) representations for dynamic scenes and introduces an optimal transport-based regularizer. Specifically, the key idea is that two images for nearby timesteps rendered from the same viewpoint should have a similar distribution over pixel values, and this is operationalized via an optimal transport-based regularization. The paper considers several different base methods and shows that this proposed regularization improves performance across these methods.

While the initial ratings diverged, the author response addressed most concerns and all the reviewers gave borderline positive ratings. The AC agrees that the proposed approach is technically interesting as well as that the paper shows is it widely applicable across base methods and datasets. Given this, this would be a beneficial paper for the community.

**Justification For Why Not Higher Score:**

While the paper is technically interesting and empirically strong, both, the technical contributions and empirical benefits are somewhat limited. This paper primarily introduces a new regularizer, which while interesting, is not something that radically improves results (but rather yields consistent but small gains).

**Justification For Why Not Lower Score:**

The proposed approach is technically interesting as well as that the paper shows is it widely applicable across base methods and datasets. Given this, this would be a beneficial paper for the community.

---

### Decision · Program_Chairs · 2024-01-16

Accept (poster)